# A Precision Assessment of a Point-of-Care Immunological Analyzer for Swift Progesterone Measurement and Guidance for Determining the Optimal Breeding Time in Bitches

**DOI:** 10.3390/ani14030377

**Published:** 2024-01-24

**Authors:** Thanikran Suwannachote, Supphathat Wutthiwitthayaphong, Saengtawan Arayatham, Wisut Prasitsuwan, Sakchai Ruenphet

**Affiliations:** 1Clinic for Small Domestic Animals and Radiology, Mahanakorn University of Technology, Bangkok 10530, Thailand; dv15006a@gmail.com; 2Immunology and Virology Department, Mahanakorn University of Technology, Bangkok 10530, Thailand; wsupphathat@mut.ac.th; 3Clinic for Gynecology and Animal Reproduction, Mahanakorn University of Technology, Bangkok 10530, Thailand; saengtawan@mut.ac.th; 4Animal Biotechnology, Mahanakorn University of Technology, Bangkok 10530, Thailand; vizutrto@gmail.com

**Keywords:** bitches, chemiluminescent microparticle immunoassay, optimal breeding time, point-of-care immunological analyzer, serum progesterone

## Abstract

**Simple Summary:**

This study seeks to address the critical role of understanding serum progesterone concentration in determining the optimal mating time. The primary objective of this research is to conduct a comparative analysis of serum progesterone results obtained from a commercial point-of-care immunological analyzer for progesterone measurement when compared to the chemiluminescent microparticle immunoassay. The overarching goal is to evaluate the accuracy of these analyzers and subsequently establish standardized guidelines for optimal breeding time in bitches. The utilization of point-of-care immunological analyzer for progesterone measurement emerges as a valuable clinical tool in the precise determination of the optimal timing for mating or artificial insemination in bitches. Furthermore, the widespread adoption of this advanced technology within the veterinary community and among breeders is expected to enhance the precision of breeding decisions, ultimately leading to significant improvements in the overall dog breeding process.

**Abstract:**

The measurement of serum progesterone often varies due to different laboratory methodologies and individual canine characteristics. In this investigation, serum progesterone outcomes obtained from a commercial point-of-care immunological analyzer, designed for efficient serum progesterone assessment in bitches, were compared with results derived from chemiluminescent microparticle immunoassay from reference laboratories in Thailand. Our thorough documentation encompassed various parameters: mean, standard deviation, 95% confidence interval, and minimum and maximum serum progesterone concentration values. Additionally, we meticulously recorded the Pearson’s correlation coefficient, Lin’s concordance correlation coefficient, and the bias correction factor. Interestingly, there was no significant difference (*p* > 0.05) in the means obtained by the point-of-care immunological analyzer and chemiluminescent microparticle immunoassay. The Pearson’s correlation coefficient between the point-of-care immunological analyzer and chemiluminescent microparticle immunoassay stood at 0.957, with Lin’s concordance correlation coefficient for point-of-care immunological analyzer recorded as 0.949. Furthermore, the bias correction factor was established at 0.991. This investigation followed established chemiluminescent microparticle immunoassay guidelines, modified to incorporate the mean and 95% confidence interval as criteria for optimal breeding time using the point-of-care immunological analyzer. In conclusion, the commercial point-of-care immunological analyzer emerges as a valuable tool, aiding in precisely determining the optimal timing for natural mating or artificial insemination in bitches.

## 1. Introduction

The determination of the optimal breeding time in bitches frequently relies on the assessment of serum progesterone (sP4) concentrations [1,2,3]. This assessment serves multiple purposes, including the detection of reproductive abnormalities such as hypoluteoidism [4,5] and the confirmation of luteolysis before parturition [6,7]. Notably, during the estrus period, a characteristic rise in sP4 concentrations beyond the baseline, frequently surpassing 1 ng/mL, marks hormonal changes. Ovulation in bitches occurs approximately 36 to 50 h following the LH peak [8], coinciding with sP4 concentrations at around 2.02 ± 0.18 ng/mL at the LH peak [9]. These concentrations then escalate to a range of 4.00 to 10.00 ng/mL on the day of ovulation [10], which presents a significant hormonal shift indicating the onset of ovulation. Intriguingly, despite this range, research by Badinand et al. [11], Seefeldt et al. [12], Marseloo et al. [13], Bouchard et al. [14], and Mir et al. [15] suggests a concentration of sP4 at 5.00–8.00 ng/mL, portraying a varied perspective on determining the ovulation date base on hormonal levels. Post-ovulation, primary oocytes persist initially, transitioning into fertilizable oocytes within 2 to 3 days [16], ultimately reaching fertilization readiness at 109 h post-ovulation [17], which is consistent with a maturation window of 96 to 108 h for oocytes within the oviduct [18]. These collective findings highlight the necessity for accurate ovulation detection to make informed decisions regarding mating or artificial insemination, underscoring its pivotal role in ensuring successful breeding outcomes.

Within the domain of veterinary practice, diverse techniques exist for quantifying sP4, including radioimmunoassay (RIA) [19,20] liquid chromatography–tandem mass spectrometry (LC-MS) [21,22,23] and chemiluminescence immunoassay (CLIA) [20,24,25,26]. Serial monitoring of sP4 concentration is crucial for predicting and confirming ovulation and detecting the fertilization period. This monitoring is intricately linked to the choice of quantification technique. Traditionally, the measurement of the sP4 concentration in bitches relied on the RIA technique, known for drawbacks such as high expense, long turnaround time, and the requirement for specialized laboratory equipment [27]. RIA, while providing quantitative values, suffers due to its reliance on specialized equipment, hindering its widespread application in monitoring sP4 for ovulation prediction. Furthermore, alternative techniques like enzyme-linked immunosorbent assay (ELISA) are available in kit form but offer only quantitative or semi-qualitative results, rendering them less reliable in precisely predicting ovulation [28]. The limitations of ELISA accentuate the necessity for more accurate and efficient in sP4 determination to optimize ovulation prediction and confirmation. An accurate and reliable alternative to address these concerns is the CLIA method, ensuring ease of assaying serial blood samples while guaranteeing safety, speed, accuracy, and repeatability [29]. CLIA not only resolves the drawbacks associated with RIA and ELISA but also provides a robust solution for precise sP4 monitoring, thereby aiding in accurate ovulation prediction and confirmation. Recent advancements, such as point-of-care sP4 measurement analyzers like rapid fluorescence immunochromatography assay (RFICA) and surface plasmon field-enhanced fluorescence spectroscopy (SPFS) [30,31,32,33], mark substantial progress in proving sP4 measurement techniques. These advancements offer new possibilities by enhancing the accuracy and accessibility of sP4 monitoring, enhancing the overall efficiency of veterinary practice in managing reproductive processes.

However, it is crucial to acknowledge that outcomes of sP4 measurement frequently exhibit variations due to the differences in laboratory methodologies and individual characteristics of bitches. Consequently, achieving precise determination of the optimal breeding time mandates the collection of numerous sequential blood samples throughout the proestrus and estrous phases. These samples function as a comparative benchmark against the gold standard or reference laboratory methodologies. With due regard to these factors, this investigation endeavors to juxtapose sP4 outcomes obtained from a commercial POC analyzer specifically developed for an expeditious sP4 assessment in bitches with those derived from chemiluminescent microparticle immunoassay (CMIA). The primary aim is to ascertain the precision and establish a standardized guideline for ascertaining the optimal breeding time.

## 2. Materials and Methods

### 2.1. Ethical Approval

The guidelines concerning the appropriate care and utilization of animals received approval from the Animal Research Ethics Committee of the Faculty of Veterinary Medicine at Mahanakorn University of Technology, Thailand. To substantiate their validation, these guidelines were identified with the specific approval code ACUC-MUT-2020/006. In accordance with the established ethical framework, the proprietors of the bitches demonstrated their concurrence to partake in the research by affixing their signatures to an official document.

### 2.2. Study Period and Location

Blood samples were procured for the purpose of analysis during the months of August 2022 and July 2023. This collection transpired at the Small Animal Teaching Hospital, Faculty of Veterinary Medicine, Mahanakorn University of Technology, Thailand, and similarly at Vet Home Polyclinic, Bangkok, Thailand.

### 2.3. Sample Collection and Progesterone Measurements

A cohort of one-hundred and ten bitches, encompassing a diverse spectrum of breeds including American bullies, English bulldog, French bulldogs, Shetland sheepdogs, Miniature American shepherds, Cavalier King Charles spaniels, Chihuahuas, Pomeranians, Chow Chows, Akitas, and Pugs, was selected for the purpose of routine estrous observation followed by sequential natural mating or artificial insemination. The research endeavor took place concurrently at the Small Animal Teaching Hospital, Faculty of Veterinary Medicine, Mahanakorn University of Technology, Thailand, and Vet Home Polyclinic, Bangkok, Thailand. Blood samples were procured from all the bitches within a time span of 5 to 7 days subsequent to the commencement of vaginal swelling or discharge. The collected blood was allowed to undergo coagulation and was subsequently subjected to centrifugation at a force of 2500× *g* for a duration of 15 min. Subsequently, the obtained sera were recovered, and two aliquots were prepared. One of these aliquots was promptly subjected to assessment for sP4 concentration through the utilization of CMIA. This process involved the application of an Architect i2000SR Immunoassay Analyzer along with the Architect Progesterone Reagent (Abbott Laboratories, Abbott park, IL, USA). The remaining aliquot was conserved at −20 °C until its need arose for assessment via a commercial POC analyzer, namely Fuji Dri-Chem Immuno AU Cartridge v-PRG (Fujifilm corporation, Tokyo, Japan), However, only 110 samples were evaluated by this POC. This process was conducted in adherence to the guidelines outlined by the manufacturer.

### 2.4. Statistical Analysis

Mean, standard deviation (SD), 95% confidence interval (CI), minimum, and maximum value for sP4 concentration were meticulously documented. Quantification was meticulously carried out employing the commercial POC analyzer and CMIA across distinct phases of the bitches’ reproductive cycle, encompassing early proestrus, LH peak, pre-ovulation, ovulation, and post-ovulation.

In the context of this investigation, the harmonization between values derived from commercial POC analyzer and those procured through the CMIA was meticulously established. This examination was executed by calculating Pearson’s correlation coefficient, Lin’s concordance correlation coefficient, and the bias correction factor. A correlation coefficient of ≤0.35 was interpreted as indicative of a low or weak correlation, while the range of 0.36–0.67 denoted a moderate correlation, 0.68–0.89 indicated a high correlation, and values exceeding 0.90 signified a very high correlation [34]. Furthermore, McBride [35] outlined a classification of agreement strength based on the Lin’s concordance correlation coefficient: >0.99 as almost perfect; 0.95–0.99 as substantial; and 0.90–0.96 as moderate; <0.90 as poor. Moreover, comprehensive Passing–Bablok regression and Bland–Altman analyses were executed for commercial POC and CMIA values. Furthermore, graphical representations were generated to vividly portray the acquired outcome. To determine the existence of a substantial distinction between the two means, a paired *t*-test was utilized. All the analyses were meticulously performed utilizing free trial version of XLSTAT in Microsoft Excel Home and Student Edition (Redmond, WA, USA) downloaded from https://www.xlstat.com/en/download (accessed on 17 January 2024). The designated significance level was set at *p* < 0.05.

## 3. Results

The individual sP4 results of the commercial POC analyzer comparison with CMIA are presented in Table 1. The analysis revealed that no significant difference (*p* > 0.05) was observed for the mean value of all samples.

Figure 1 presents a visual representation of the Passing–Bablok regression plot, illustrating a comparison between serum progesterone measurements acquired from the POC analyzer and those derived from CMIA. The associated regression equations and correlation coefficient (r) are provided as follows: y = 1.114x − 0.597. The analysis of Bland–Altman plots is shown in Figure 2. The mean (range) biases between the POC analyzer and CMIA were 0.22 ng/mL (−5.24 to 5.69 ng/mL).

The means, SD, 95% CI, and range of sP4 concentrations—as determined with the CMIA and POC analyzers during the early proestrus, LH peak, pre-ovulation, ovulation and post-ovulation periods of the bitch—are presented in Table 3.

## 4. Discussion

Several researchers opt for employing reference standards such as RIA [20,36] or LC-MS [37] for sP4 measurement in bitches. However, the inconvenience associated with RIA, stemming from the requirement for specialized laboratory equipment—particularly radioactive substances posing risks to both personnel and the environment—had led to a shift. Presently, CLIA is increasingly displacing RIA in numerous diagnostic laboratories, and is becoming the preferred method for quantifying progesterone among veterinary practitioners [38,39]. The evolution to CMIA, a more advanced iteration of CLIA, implemented in veterinary reference laboratories throughout Thailand [30], signifies the progression in sP4 measurement techniques. Nevertheless, in certain regions outside Thailand, including remote areas, CLIA faces availability challenges, necessitating time for sample transportation and leading to prolonged result turnaround times. This context highlights the practical challenges of CLIA, implemented in veterinary reference laboratories throughout Thailand [30], signifies the progression in sP4 measurement techniques.

The expeditious determination of sP4 concentrations holds pivotal significance in facilitating precise diagnoses and well-informed clinical decisions, especially in contexts such as mating or the meticulous management of cesarean sections [40]. Establishing historical context, the gold standard for determining sP4 concentrations in bitches was the Coat-A-Count 125I RIA by Siemens, which was discontinued in 2014, leading to a need for newer validated methods like the CLIA [41]. Since different assays utilize distinct technologies, inherent differences in the analyte levels reported by each assay can occur, prompting the introduction of newer validated methods like the CLIA, such as IMMULITE by Siemens, which has gained wide acceptance in research and clinical practice [9,24,42,43]. However, while similar to RIA, CLIA is performed by commercial diagnostic laboratories, and its extended turnaround time accentuates the necessity for an improved approach towards faster and more efficient diagnostic methods. Consequently, veterinary practitioners consider the adoption of a commercial POC immunologic analyzer indispensable. Addressing this need, this study evaluated an automated fluorescence immunoassay analyzer, which showcases a rapid processing time and likely portability, directly addressing the limitations faced in the current methodologies. Moreover, the introduction of alternative POC analyzers like Catalyst^®^ Progesterone, which provides quicker results, significantly enhances the diagnostic potential, especially in late-term pregnant female dogs [24,44].

The correlation coefficient, often referred to as Pearson’s product–moment *r*, requires both magnitude and direction—either positive or negative. It signifies a range from −1 to +1, denoting absolute and nondimensional values with no association between the variables measured, emphasizing the relationship between magnitude and association. A correlation coefficient of zero indicates no association between the measured variables. The strength of the coefficient remains independent of its direction or sign, offering a clear interpretation of the relationship. Thus, as the *r* coefficient approaches ±1, association between variables becomes stronger, indicating a more linear relationship. A positive correlation implies an increase in one variable corresponding to an increase in the other, suggesting a direct relationship. Conversely, a negative correlation denotes an inverse relationship: as one variable increases, the other decreases [34]. In this study, the correlation analysis between a commercial POC analyzer and CMIA revealed Pearsons’ correlation coefficients exceeding the significant threshold of 0.90 (Table 2), demonstrating robust and meaningful associations. However, variations in the assessment of the strength of agreement, evaluated through Lin’s concordance correlation coefficient, were observed in the analyzers at a substantial level [35].

In this study, the serum was handled differently between the two different analyzers, with half of the harvested serum analyzed using the CMIA analyzer, and the remaining half frozen and stored at −20 °C prior to POC analyzer testing. Bolelli et al. [45] reported that the progesterone level decreased after long-term storage at −70 °C, either due to molecular modification or due to interference by the cryotube material used in low-temperature storage. On the other hand, apparently normal progesterone levels in sera stored at −20 °C for up to 7 years were reported [46]. Volkmann’s study in 2006 [43] explored the impact of anticoagulant, storage time, temperature, and assay methods on blood progesterone concentrations in dogs. The outcomes of the study revealed several key findings: (i) the RIA measurements yielded a significantly higher sP4 concentration than CLIA; (ii) refrigeration of whole blood during the initial 2 h post-sample collection notably decreased the measured sP4 concentration; (iii) the progesterone concentration in heparinized plasma appeared to be unaffected by the storage temperature of whole blood for at least 5 h; and (iv) the refrigeration of whole, clotted blood did not impact the sP4 concentration, provided that the samples were held at room temperature for the first 2 h after collection. To enhance coherence, these findings align with the primary focus on understanding the effects of various factors on blood progesterone levels in canine subjects. Therefore, in this study, all samples were promptly separated within 2 h after whole blood collection to obtain serum. This rapid separation and subsequent storage at −20 °C were conducted to maintain sample integrity and stability prior to POC immunological analyzer testing, ensuring accurate analysis and minimizing any potential degradation of analytes.

Görlinger et al. [47] reported insufficient progesterone secretion by the corpora lutea during pregnancy in a Bernese Mountain dog, specifically termed hypoluteoidism. To establish a diagnosis of hypoluteoidism accurately, plasma progesterone concentration measurement, performed with an RIA, is crucial. Commercially available ELISA kits lack reliability in the critical range of 0.87–4.61 ng/mL. Therefore, accurate measurement methods are necessary. Additionally, it is imperative that the RIA used does not detect exogenous progesterone sources used for preventing abortion. This exclusion of detecting medroxyprogesterone acetate in the RIA allows researchers to focus on monitoring endogenous plasma progesterone levels effectively. In our study, CMIA and POC analyzers were employed to monitor pregnant bitches treated with medroxyprogesterone acetate. Importantly, this study did not detect exogenous progesterone sources until parturition.

Table 3 presents the mean, SD, 95% CI, and range for serum progesterone concentration with a quantification comparison between the commercial POC analyzer and CMIA for estimates during the early proestrus, LH peak, pre-ovulation, ovulation, and post-ovulation periods of the bitches. This table displays discernible sets of values, denoted by lowercase letters, which are meticulously organized in the respective rows for POC analyzer. Each of these values, associated with a distinct superscript, demonstrates a statistically significant difference (*p* < 0.05) when compared to the reference CMIA values. Conversely, the uppercase lettered values, presented similarly with their own set of superscripts, demonstrate no statistically significant difference *(p* > 0.05). The absence of a significant difference indicates that these results are comparable to the CMIA results. Importantly, the mean for all periods in the POC analyzer revealed no difference from CMIA. However, the ovulation period of the meticulously estrous cycle using the POC analyzer showed no significant difference.

The mean, 95% CI, and minimum/maximum of the sP4 concentrations obtained at the estimated time of ovulation in this study are of particular significance due to their potential implications in reproductive management. Moreover, considerable variations in sP4 concentrations were observed, depending on the factor, that could potentially lead to confusion among veterinary professionals. For instance, when a single plasma sample was divided into seven aliquots and subjected to sP4 concentration assays using various techniques across seven independent laboratories, resulting mean sP4 concentrations were as follows: 4.6, 3.6, 6.8, 7.2, 3.9, 9.2, and 5.2 ng/mL [48]. Recent investigations by Tahir et al. [49], and Schmicke et al. [39] underscore the substantial variations in sP4 concentrations during the estimated time of ovulation, contingent upon the assay employed. However, the sP4 concentrations determined using CMIA at the estimated time of ovulation in this study (mean ± SD = 7.05 ± 1.41 ng/mL; 95% CI = 6.69 to 7.40 ng/mL; range = 5.07 to 9.78 ng/mL) displayed no statistically significant difference compared to this POC analyzer (mean ± SD = 6.14 ± 2.03 ng/mL; 95% CI = 5.37 to 6.92 ng/mL; range = 2.24 to 10.35 ng/mL). Consequently, our findings hold the potential to guide optimal breeding time management, with implications based on the 95% CI and range (minimum-maximum) of sP4 results (Table 4). Furthermore, this research suggests avenues for further studies, particularly in the realm of investigating specific refinements to the assessment of sP4. These refinements aim to bolster veterinary practice and enhance reproductive management by providing more precise and reliable tools. In alignment with this objective, the guidelines presented herein were meticulously crafted. They were developed by being aligned with the established CMIA guidelines and were adapted using the range and 95% CI derived from each set of results (Table 4).

In this study, the bitches did not only determine the progesterone concentration but also underwent a vaginal cytology examination to ensure that the optimal breeding time was predicted correctly. On the first day when vaginal cytology indicated ≥70% cornified epithelial cells, the bitch was considered in late proestrus, and a blood sample for progesterone determination was collected. On the day when vaginal cytology showed ≥90% cornified epithelial cells, the bitch was considered in estrus, and blood samples were collected following the guideline in Table 4. Furthermore, this investigation aimed to comprehensively assess 63 bitches to determine their ovulation day and predict their optimal breeding dates. This involved employing both CMIA and POC analysis methods during proestrus and estrus periods before breeding, aimed at refining prediction accuracy. At or near the ovulation phase, artificial insemination was performed within two to four days, aligning with the determined optimal breeding window. The results indicated a promising outcome. In total, 55 out of 63 bitches successfully conceived, resulting an 87.30% pregnancy rate, while 52 out of 63 achieved parturitions, indicating an 82.54% parturition rate, confirming the precision of the predicted breeding window. Despite this, the absence of significant differences among the pregnant and non-pregnant bitches of the studied breeds presents a conflicting finding that requires further investigation. These findings underscore the essential role of CMIA and POC analyzers in improving breeding and parturition outcomes. The rapid decision making enabled by the POC analyzer benefits both veterinary practitioners and breeders, facilitating more precise and timely breeding decisions aligned with optimal fertility windows, thereby enhancing future breeding success based on the investigation’s results.

## 5. Conclusions

The utilization of a commercial POC immunologic analyzer specifically designed for progesterone measurement emerges as a critical clinical tool for determining the precise timing of natural mating or artificial insemination in bitches. In this study, employing this analyzer revealed a pregnancy rate of at least 87%, adhering meticulously to the standardized guidelines. This finding underscores the analyzer’s efficacy in predicting successful optimal breeding times, emphasizing its role in reproductive management. Moreover, the broader adoption of this advanced technology within the veterinary community and among breeders has the potential to significantly enhance the quality of breeding decisions. Leveraging the success observed in this study, a widespread integration of this technology could markedly improve overall dog breeding processes. However, further empirical research is required to validate this potential, especially in a variety of dog breeds such as small, medium, and large breeds.

## Figures and Tables

**Figure 1 animals-14-00377-f001:**
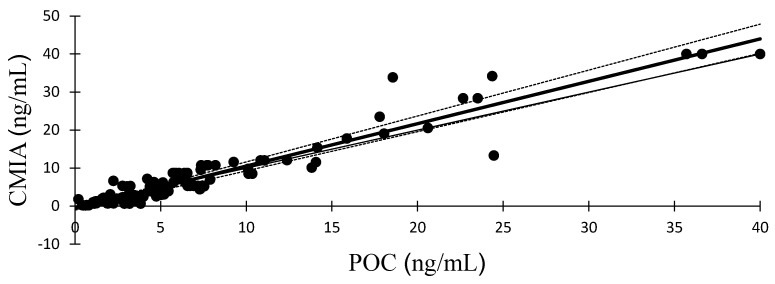
Passing–Bablok regression plot depicting serum progesterone concentration (*n* = 110) determined with commercial point-of-care (POC) immunological analyzer and chemiluminescence microparticle immunoassay (CMIA). The thin line is the identity line. The thick line represents the regression line (y = 1.114x − 0.597; r = 0.957) and the dotted lines represent its 95% confidence interval.

**Figure 2 animals-14-00377-f002:**
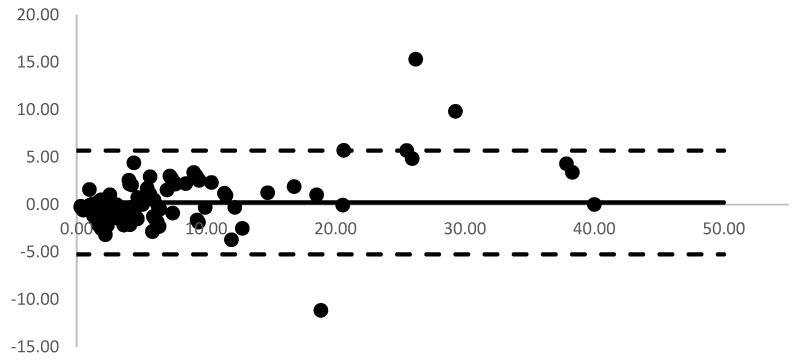
Bland–Altman plot of agreement between 110 pairs of serum progesterone concentrations determined with commercial point-of-care (POC) immunological analyzer and chemiluminescence microparticle immunoassay (CMIA). The black thick line represents the mean difference (bias) and the black dotted line represents the 95% confidence interval.

**Table 1 animals-14-00377-t001:** Serum progesterone concentration of the chemiluminescence microparticle immunoassay (CMIA) and commercial point-of-care (POC) immunological analyzer for individual bitches (*n* = 110).

Bitch	CMIA (ng/mL)	POC (ng/mL)	Bitch	CMIA (ng/mL)	POC (ng/mL)	Bitch	CMIA (ng/mL)	POC (ng/mL)	Bitch	CMIA (ng/mL)	POC (ng/mL)
1	0.20	0.47	31	1.98	2.44	61	5.22	7.54	91	10.75	7.37
2	0.20	0.61	32	1.98	1.82	62	5.29	3.23	92	11.55	14.07
3	0.20	0.62	33	1.98	1.59	63	5.31	6.57	93	11.58	9.26
4	0.22	0.74	34	1.98	2.59	64	5.31	7.02	94	12.02	10.83
5	0.22	0.81	35	1.98	1.80	65	5.31	6.94	95	12.02	11.06
6	0.22	0.66	36	2.17	1.66	66	5.31	7.17	96	12.08	12.38
7	0.24	0.42	37	2.17	1.77	67	5.31	6.71	97	13.30	24.45
8	0.26	0.69	38	2.33	3.00	68	5.33	2.77	98	15.39	14.15
9	0.64	3.18	39	2.33	2.78	69	6.20	6.43	99	17.76	15.87
10	0.64	2.88	40	2.51	4.00	70	6.20	6.69	100	19.05	18.04
11	0.64	3.83	41	2.57	4.74	71	6.21	5.13	101	20.54	20.60
12	0.66	1.12	42	2.85	3.37	72	6.22	5.75	102	23.50	17.79
13	0.73	1.93	43	2.85	3.48	73	6.31	4.61	103	28.35	22.66
14	0.73	2.25	44	2.98	5.04	74	6.63	2.24	104	28.35	23.51
15	0.73	1.88	45	3.00	4.75	75	6.98	7.89	105	33.86	18.55
16	0.74	1.25	46	3.00	5.07	76	7.15	4.22	105	34.18	24.36
17	1.00	1.06	47	3.07	5.20	77	7.75	6.22	107	40.00	40.00
18	1.03	2.00	48	3.09	2.06	78	8.49	10.35	108	40.00	35.70
19	1.16	1.70	49	3.10	3.14	79	8.50	10.13	109	40.00	36.61
20	1.19	1.38	50	3.88	4.75	80	8.70	6.07	110	40.00	40.00
21	1.20	1.26	51	3.88	4.45	81	8.71	6.56			
22	1.23	1.30	52	3.94	5.36	82	8.71	5.88			
23	1.24	1.26	53	3.94	5.46	83	8.71	5.71			
24	1.24	1.20	54	3.95	5.03	84	8.71	6.38			
25	1.24	1.27	55	4.05	4.34	85	9.54	7.34			
26	1.28	3.49	56	4.44	7.28	86	9.78	10.10			
27	1.28	3.42	57	4.44	7.30	87	10.11	13.82			
28	1.30	1.40	58	5.09	5.10	88	10.74	7.70			
29	1.79	0.20	59	5.09	4.37	89	10.74	7.76			
30	1.85	3.35	60	5.18	3.03	90	10.74	8.20			

**Table 2 animals-14-00377-t002:** Measure of agreement: concordance correlation coefficient, Pearsons’ correlation coefficient ant bias correction factor (*n* = 110).

95% Limits of Agreement (Bland and Altman)			Concordance Correlation Coefficient	95% Confidence Interval		Pearsons’ Correlation Coefficient	Bias Correction Factor
Average Difference	Lower	Upper		Lower	Upper		
0.22	−5.24	5.69	0.949	0.929	0.963	0.957	0.991

**Table 3 animals-14-00377-t003:** Mean, standard deviation (SD), 95% confidence interval (CI), minimum (Min), and maximum (Max) values for serum progesterone concentration with quantification using the chemiluminescence microparticle immunoassay (CMIA) and commercial point-of-care (POC) immunological analyzer for estimates during the proestrus, LH peak, pre-ovulation, ovulation, post-ovulation, and all periods of the bitches.

	CMIA (ng/mL)			POC (ng/mL)		
Period (*n*)	Mean ± SD	95% CI	Min–Max	Mean ± SD	95% CI	Min–Max
Proestrus (35)	1.00 ± 0.60 ^A^	0.80–1.21	0.20–1.98	1.65 ± 0.98 ^a^	1.32–1.99	0.20–3.83
LH peak (9)	2.53 ± 0.31 ^B^	2.29–2.76	2.17–2.98	3.32 ± 1.17 ^b^	2.41–4.22	1.66–5.04
Pre-ovulation (13)	3.67 ± 0.54 ^C^	3.35–4.00	3.00–4.44	4.94 ± 1.41 ^c^	4.09–5.79	2.06–7.30
Ovulation (29)	6.80 ± 1.57 ^D^	6.20–7.40	5.09–9.78	6.14 ± 2.03 ^D^	5.37–6.92	2.24–10.35
Post-ovulation (24)	21.11 ± 11.35 ^E^	16.32–25.90	10.11–40.00	18.95 ± 10.27 ^e^	14.61–23.29	7.37–40.00
All period (110)	7.36 ± 9.28 ^F^	5.61–9.11	0.20–40.00	7.13 ± 8.15 ^F^	5.59–8.67	0.20–40.00

^a,b,c,e^ values within the rows of POC with different superscripts represent statistically significant differences (*p* < 0.05) when compared with CMIA, while ^A,B,C,D,E,F^ values are show no significant differences (*p* > 0.05).

**Table 4 animals-14-00377-t004:** Reference or guideline for serum progesterone interpretation using commercial point-of-care (POC) immunological analyzer in heat or apparent reproductively quiescent bitches.

Progesterone by CMIA (ng/mL)	Progesterone by POC (ng/mL)	Likely Events	Suggestion
	Min–Max (95% Confidence Interval)		
<2	0.20–3.83 (1.32–1.99)	Anestrus, proestrus, and pre-LH surge	- Confirm heat or proestrus by physical examination or vaginal cytology - Retest in 3 days.
2.00–2.99	1.66–5.04 (2.41–4.22)	LH surge	- Retest in 2 days to confirm continued rise in progesterone. - Aim for breeding 4–7 days.
3.00–4.99	2.06–7.30 (4.09–5.79)	Pre-ovulation	- Retest in 1–2 days to confirm continued rise in progesterone. - Aim for breeding 3–5 days.
5.00–9.99	2.24–10.35 (5.37–6.92)	At or near ovulation	- Retest in 1 day to confirm continued rise in progesterone. - Aim for breeding 2–4 days.
>10	7.37–40.00 (14.61–23.29)	Post-ovulation, oocyte maturation, and in fertilizable period	- Aim for breeding on this day and for another 2 days hereafter.

## Data Availability

The data presented in this study are available free of charge for any user on request from the corresponding authors.

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
