# Peer review of "A Precision Assessment of a Point-of-Care Immunological Analyzer for Swift Progesterone Measurement and Guidance for Determining the Optimal Breeding Time in Bitches"

_animals, 2024, doi:10.3390/ani14030377_

Round 1
Reviewer 1 Report
Comments and Suggestions for Authors
The authors are testing the efficacy of the utilization of a commercial point-of-care (POC) immunological analyzer for the accurate measurement of serum progesterone in female dogs. To evaluate the POC analyzer the authors compare the output data to the data obtained from the traditional chemiluminescent microparticle immunoassay (CMIA) analyzer. The authors claim that the samples analyzed by the 2 different analyzer did not measure significant differences, concluding that the POC can be used to measure progesterone in canine serum. However, the serum was handled differently between the use of the 2 different analyzers. Half of the harvested serum was analyzed using the CMIA analyzer and the remaining half was frozen and stored at -20degrees celsius. The authors need to include reference controls where known quantities of progesterone are added to canine serum, fresh and frozen samples that do not have endogenous native progesterone, and analyzed using the CMIA and POC to validate measurement of known added quantities.
Author Response
Comments: The authors are testing the efficacy of the utilization of a commercial point-of-care (POC) immunological analyzer for the accurate measurement of serum progesterone in female dogs. To evaluate the POC analyzer the authors compare the output data to the data obtained from the traditional chemiluminescent microparticle immunoassay (CMIA) analyzer. The authors claim that the samples analyzed by the 2 different analyzers did not measure significant differences, concluding that the POC can be used to measure progesterone in canine serum. However, the serum was handled differently between the use of the 2 different analyzers. Half of the harvested serum was analyzed using the CMIA analyzer and the remaining half was frozen and stored at -20 degrees celsius. The authors need to include reference controls where known quantities of progesterone are added to canine serum, fresh and frozen samples that do not have endogenous native progesterone, and analyzed using the CMIA and POC to validate measurement of known added quantities.
Response: We described references that confirmed frozen samples do not changed the level of progesterone (Line 236-254). Moreover, we explored the limitations of exogeneous progesterone (Line 265-274).
Reviewer 2 Report
Comments and Suggestions for Authors
Dear authors,
I have reviewed your article and have only few comments regarding Table 3:
1) Values for "Ovulation" and "post-ovulation" are the same in columns "Mean+-SD" and "95%CI". It is evidently a mistake. Similarly, there are same values for "LH Peak" in columns "95%CI" and "Min-Max". This also means, that you should check that data presented in row "All Period" are all correct.
2) I miss the definition of the values in the table 3? Are they ng/ml? It needs to be defined.
3) Moreover, in the same table you refer to notes "a,b,c,d,e,f" but "d,f" are not present in the table.
Author Response
Comments: I have reviewed your article and have only few comments regarding Table 3:
- Values for "Ovulation" and "post-ovulation" are the same in columns "Mean+-SD" and "95%CI". It is evidently a mistake. Similarly, there are same values for "LH Peak" in columns "95%CI" and "Min-Max". This also means, that you should check that data presented in row "All Period" are all correct.
Response: Rechecked Table-3 with raw data already (Line 261-262).
Comments: 2) I miss the definition of the values in the table 3? Are they ng/ml? It needs to be defined.
Response: Already added to Table-3 (Line 261-262).
Comments: 3) Moreover, in the same table you refer to notes "a,b,c,d,e,f" but "d,f" are not present in the table.
Response: Already corrected below Table-3 (Line 262-264).
Reviewer 3 Report
Comments and Suggestions for Authors
The present study seeks to address the critical role of understanding serum progesterone (sP4) concentration in determining the optimal mating time. The primary objective of this research is to conduct a comparative analysis of sP4 results obtained from a commercial point-of-care (POC) immunological analyzer for progesterone measurement when compared to the chemiluminescent microparticle immunoassay (CMIA). A major revision is required to address the appended concerns:
Line 27-40: All abbreviations in the abstract section should be fully stated.
Line 61-76: please try to improve the rationale of the study. It is not clear whether this purpose was not investigated before or not. Try to state the previous literature about the measurement of canine progesterone as an indicator for ovulation.
Line 91-109: Did the authors find any significant differences among the studied breeds?
Line 104-109: Did the authors have any information about the sensitivity and specificity of the assay?
Line 110: No information about the normality and distribution of data. Please justify.
Line 142-150: please state here the clinical significance of the findings of this study.
Line 151-165: Please support your statement with relevant references.
Line 166-170: Please explain this strong correlation.
In all tables, please indicate the sample number.
Line 254-259: How the authors guarantee this conclusion. The authors did not assess the fertility potential. What is the gold standard method to reach this conclusion?
Comments on the Quality of English Language
Minor editing of English language required
Author Response
The present study seeks to address the critical role of understanding serum progesterone (sP4) concentration in determining the optimal mating time. The primary objective of this research is to conduct a comparative analysis of sP4 results obtained from a commercial point-of-care (POC) immunological analyzer for progesterone measurement when compared to the chemiluminescent microparticle immunoassay (CMIA). A major revision is required to address the appended concerns:
Comments: Line 27-40: All abbreviations in the abstract section should be fully stated.
Response: Corrected in the simple summary and abstract following your suggestion (Line 16-45).
Comments: Line 61-76: please try to improve the rationale of the study. It is not clear whether this purpose was not investigated before or not. Try to state the previous literature about the measurement of canine progesterone as an indicator for ovulation.
Response: Corrected in the introduction following your suggestion (Line 50-66).
Comments: Line 91-109: Did the authors find any significant differences among the studied breeds?
Response: Added this sentence in discussion “The results indicated a promising outcome: 55 out of 63 bitches successfully conceived, resulting an 87.30% pregnancy rate, while 52 out of 63 achieved parturitions, indicating an 82.54% parturition rate, confirming the precision of the predicted breeding window. Despite this, the absence of significant differences among the pregnant and non-pregnant bitches of the studied breeds presents a conflicting finding that requires further investigation.” (Line 315-320)
Comments: Line 104-109: Did the authors have any information about the sensitivity and specificity of the assay?
Response: Generally, sensitivity and specificity were utilized for quality assessment, intending to distinguish results as positive or negative and calculate these metrics for both assays. However, given the quantitative nature of the present study's assessment, the outcomes couldn't be reported as negative or positive. As a result, these parameters were assessed using correlation coefficients or Passing-Bablok between both assays. This allows for a more nuanced evaluation considering the quantitative nature of the data, ensuring a comprehensive understanding of the assays' performance and agreement.
Do you have any comment about this point? Could you advise me.
Comments: Line 110: No information about the normality and distribution of data. Please justify.
Response: In Table 3, I just added the number of samples (n) in each period of the present study. However, the distribution represented in Figure 1 and Figure 2 reflects irregularities. These irregularities arise from the nature of serum progesterone examination, aimed at observing the ovulation day, resulting in higher density numbers before ovulation compared to during and after the ovulation periods. This explanation clarifies the irregularities observed in the distribution across the different study periods.
Comments: Line 142-150: please state here the clinical significance of the findings of this study.
Response: In the discussion section, we elaborated more on the clinical significance of POC (Line 178-199).
Comments: Line 151-165: Please support your statement with relevant references.
Response: Added and corrected this paragraph following your suggestion (Line 183-199).
Comments: Line 166-170: Please explain this strong correlation.
Response: We elaborated more on the correlation coefficient following your suggestion (Line 205-218).
Comments: In all tables, please indicate the sample number.
Response: Corrected followed your suggestion.
Comments: Line 254-259: How the authors guarantee this conclusion. The authors did not assess the fertility potential. What is the gold standard method to reach this conclusion?
Response: We elaborated more on the conclusion section (Line 326-335).
Round 2
Reviewer 1 Report
Comments and Suggestions for Authors
The authors have adequately addressed concerns.
Author Response
Comments: The authors have adequately addressed concerns.
Response: Thank you very much
Reviewer 3 Report
Comments and Suggestions for Authors
Thank you very much for addressing comments in the first round of reviewing. The manuscript is improved. Few issues should be fixed:
1-line 266-275: Can the authors discuss/recommend using of progesterone assay altogether with other clinical investigations/approaches such as vaginal smear cytology to enhance the diagnostic value of progesterone assay? Please discuss this issue.
2-Line 333-336: Please justify what should you recommend to do research in future to investigate this issue?
3-Moderate English revision is needed.
Comments on the Quality of English LanguageModerate English revision is needed.
Author Response
Thank you very much for addressing comments in the first round of reviewing. The manuscript is improved. Few issues should be fixed:
Comments: 1-line 266-275: Can the authors discuss/recommend using of progesterone assay altogether with other clinical investigations/approaches such as vaginal smear cytology to enhance the diagnostic value of progesterone assay? Please discuss this issue.
Response: Corrected in the discussion following your suggestion (Line 311-316).
Comments: 2-Line 333-336: Please justify what should you recommend to do research in future to investigate this issue?
Response: Corrected in the discussion following your suggestion (Line 339-341).
Comments: 3-Moderate English revision is needed.
Response: Already checked by English native speaker.